Evaluation of animal control measures on pet demographics in Santa Clara County, California, 1993–2006

Kass Philip H. phkass@ucdavis.edu 1
Johnson Karen L. 2
Weng Hsin-Yi 3
1 Department of Population Health and Reproduction, School of Veterinary Medicine, University of California , Davis, CA , USA
2 National Pet Alliance , San Jose, CA , USA
3 Department of Comparative Pathobiology, College of Veterinary Medicine, Purdue University , West Lafayette, IN , USA
Perencevich Eli
Electronic publication date: 2013 Feb 19
Publication date: 2013
Volume: 1
Electronic Location ID: e18
Received 2012 Nov 14; Accepted 2013 Jan 4
Copyright: © 2013 Kass et al.
Copyright year: 2013
Copyright holder: Kass et al.
License: This is an open access article distributed under the terms of the Creative Commons Attribution License, which permits unrestricted use, distribution, and reproduction in any medium, provided the original author and source are credited.
License URL: https://creativecommons.org/licenses/by/3.0/

Keywords: Animal euthanasia, Epidemiology, Animal population groups, Population control, Population policy

Funding: The Cat Fanciers’ Association, Inc Funding for this research was provided by The Cat Fanciers’ Association, Inc.; The George and Phyllis Miller Trust, Center for Companion Animal Health, School of Veterinary Medicine, University of California; City of San Jose; County of Santa Clara; National Pet Alliance; Tails and No Tales Cat Club; Santa Clara Valley Cat Fanciers; Ft. Worth Cat Club, and numerous anonymous individuals. The funders had no role in study design, data collection and analysis, decision to publish, or preparation of the manuscript.

==============================
The measurable benefits of animal control programs are unknown and the aim of this study was to determine the impact of these programs on pet population changes. A prospective cross-sectional study of 1000 households was implemented in 2005 to evaluate characteristics of the owned and unowned population of dogs and cats in Santa Clara County, California. The same population was previously studied 12 years earlier. During this time period, the county instituted in 1994 and then subsequently disestablished a municipal spay/neuter voucher program for cats. Dog intakes declined from 1992–2005, as they similarly did for an adjacent county (San Mateo). However, cat intakes declined significantly more in Santa Clara County than San Mateo, with an average annual decline of approximately 700 cats for the 12 year period. Time series analysis showed a greater than expected decline in the number of cats surrendered to shelters in Santa Clara County during the years the voucher program was in effect (1994–2005). The net savings to the county by reducing the number of cat shelter intakes was estimated at approximately $1.5 million. The measurable benefits of animal control programs are unknown and the aim of this study was to determine the impact of these programs on pet population changes.

One of the greatest threats to the lives of cats and dogs in contemporary American society does not come from infectious or noninfectious disease, but rather from the threat of being unowned or becoming unwanted and susceptible to abandonment or relinquishment to shelters. Each year millions of healthy and potentially adoptable pets are euthanized for lack of ownership or residence; the most palpable manifestation of this is witnessed at local municipal or private animal shelters (Salman et al., 1998). The financial burden of managing this overabundance of pets to communities across the United States is enormous and incalculable (Zanowski, 2010).

Scientific investigations into pet population dynamics have evolved from the purely descriptive to the analytic, particularly with respect to studying determinants of relinquishment. Many studies performed in the United States have sought to quantify characteristics of animals as well as their owners that appear to be proscriptive for an impairment of the human–animal bond (Patronek et al., 1996a; Patronek et al., 1996b; Salman et al., 1998; New et al., 1999; Scarlett et al., 1999; Kass et al., 2001). The cumulative effect of these studies has led to a better understanding of why relinquishment occurs, but the enduring challenge remains how to use such information to implement prevention and/or intervention strategies. A prototypical example of these strategies is the establishment of community spay and neuter programs. Such programs can be sponsored either by municipalities or humane organizations, both of which often jointly serve critical animal control needs in communities and frequently join together in collaboration to achieve their mutual goals.

Santa Clara County, California is an opportune place to study the results of intervention strategies. As of 2005 the 1,291 square mile county contained 1.76 million people (in an estimated 603,000 households, averaging 2.92 persons per household), with more than half (945,000) living in San Jose, and over 200,000 more living in the cities of Sunnyvale and Santa Clara (US Census Bureau, 2012; State of California, 2009a). The ethnic distribution was approximately 44% Caucasian, 25% Asian/Pacific Islander, 24% Hispanic/Latino, 3% African-American, and 4% other groups (State of California, 2009b). Per capita annual growth has been approximately 1.2%; annual household growth has increased approximately 0.8% over the past 15 years (US Census Bureau, 2012; State of California, 2009a).

Two major animal shelters operate: the City of San Jose Animal Care & Services (SJACS) which opened in 2004, and the Humane Society of Silicon Valley (HSSV) facility in Santa Clara, which accepted up to 25,000 animals per year. The latter predominantly provided sheltering services until late 1992, when for financial reasons the County ceased field services for cats, only accepting those owner-reliquished. The HSSV recommenced services 14 months later for most of the County.

The number of cats entering the HSSV climbed approximately 25% (from 20,000 to 25,000 cats) from 1983 to 1990, and remained close to its high until Santa Clara County field services ended in 1992; in 1993 the total number of incoming live cats returned to 20,000 (Cat Fanciers’ Almanac, 1994). Approximately 60% of incoming animals to HSSV arrived through field services. Upon resumption of these services in 1994, San Jose instituted a free spay/neuter voucher program to reduce its number of stray cats, and initiated one for dogs in 1995. These programs ceased in 2003 in anticipation of the SJACS opening, but temporarily resumed in 2005 until the latter opened its own low-cost spay/neuter clinic in 2006. Santa Clara County also instituted a low-cost spay/neuter program in 1998 with almost $50,000 in annual funding.

A local non-profit organization of cat and dog owners and fanciersa commissioned a survey of Santa Clara County residents in 1993 (Cat Fanciers’ Almanac, 1994). The purpose of this survey was estimate the number of owned and unowned cats and dogs in the county. Investigators interviewed people by telephone from 1,031 households throughout all parts of the County except the city of Palo Alto (whose small shelter declined to provide intake statistics), and determined that 51% of households did not own pets, 19% owned only cats, 19% owned only dogs, and 11% owned both cats and dogs. Households that owned cats and dogs had an average of 1.7 cats and of 1.3 dogs, respectively. Ten percent of all households (48% of which did not own pets) also cared for an average of 3.4 stray cats. Strays, sometimes referred to as community cats, are free-roaming, unowned, or feral; the latter do not allow human touch. These figures led investigators to estimate that in 1993 the number of owned cats and dogs in the county was approximately 247,000 and 195,000, respectively (Cat Fanciers’ Almanac, 1994). Of particular importance was the projection that the County had approximately 169,000 unowned but fed cats (41% of all cats in the county), a figure that does not account for unowned but unfed and feral cats.

The purpose of the current study, conducted in 2005, was to revisit the population of cats and dogs in Santa Clara County 12 years following the earlier study, and to correlate pet population changes with the institution or disestablishment of animal population control programs, including vouchers, field services, and low-cost spay and neuter facilities. The hypothesis was that these programs would be associated with a reduction in the pet populations in Santa Clara County that differed from that of an adjacent comparison county, with resultant cost savings to the county.

Materials and Methods

Study population

The same private survey firm used in 1993 was commissioned to conduct a similar random telephone survey of 1,000 households throughout Santa Clara County except Palo Alto (which has its own small shelter and did not participate in 1993). An equal probability of selection method (EPSEM) phone list of residential landline telephone numbers for the survey was purchased from a private company.4 Three attempts were made at each number over successive nights. Over 7,000 calls (including disconnected lines, no answers, refusals) were attempted to reach 1000 respondents. People who agreed to be questioned were asked whether or not they owned dogs or cats, fed stray dogs or cats, whether or not the animals had been altered, if they had reproduced, if cats had been declawed, how they obtained their pets, whether or not cats were allowed outside, purebred status, city of residence, and residence type. Data was initially recorded on written interview forms, and manually entered into a Microsoft Excel 2007 (Microsoft Corporation, Redmond, WA) spreadsheet for statistical analysis.

Animal shelter entry information was provided by Santa Clara County Animal Control, Humane Society of Silicon Valley, San Jose Animal Care and Services, and for comparative purposes the Peninsula Humane Society and SPCA in neighboring San Mateo County and Los Angeles County Animal Control. Spay/neuter voucher program information, costs and statistics were obtained from the City of San Jose, and County of Santa Clara. San Mateo County was chosen for comparative purposes, as it most closely resembled Santa Clara County, as opposed to the other four more rural surrounding counties.

Statistical analysis

The 1982–1993 Santa Clara County shelter intake records (from before the launch of the spay/neuter voucher program) were used for projecting the expected numbers of shelter intakes from 1994 to 2005. US Census data was used to determine the number of county households, which was used with survey-derived estimates of the average number of dogs and cats per household and proportion of households that owned dogs and cats to estimate the number of owned dogs and cats in the county, and with survey-derived estimates of the percentage of households feeding stray dogs and the average number of unowned but fed dogs and cats per household to estimate the number of stray dogs and cats in the county.

Data from the survey was initially described using proportions. Pearson’s chi-square test was used to compare proportions; p-values < 0.05 were considered statistically significant. The observed numbers of shelter intakes between 1994 and 2005, during which the spay/neuter program was in place, were compared with projected numbers based on varying the proportion of cats in the voucher program that were owned versus unowned to assess the program’s effect (i.e., change in numbers of shelter intakes). Autoregressive integrated moving average (ARIMA (p, q, d, where p =  order of autoregression, q =  order of moving averages, and d =  order of differencing)) models were fit to the data before the launch of the spay/neuter voucher program (1982 to 1993). Autocorrelation function (ACF) and partial autocorrelation function (PACF) plots were used to select the best-fit ARIMA models and to evaluate the model fit. The selected ARIMA models were then used to estimate and project the trend in number of shelter intakes after the implementation of spay/neuter program (1994 to 2005) with corresponding 95% confidence limits of the ARIMA projections. In addition, we constructed a stochastic model to estimate the number of additional cats that would have been born and taken into the shelters between 1994 and 2005 had the spay/neuter program never been implemented. The key parameters, their corresponding probability distributions for owned and unowned cats, and the data sources are listed in Table 1. This model was also used for benefit–cost analysis of the voucher program. The software program @Risk (version 5.0.0, Palisade Corp., Ithaca, New York) was used for the simulation, using Latin Hypercube sampling and Mersenne Twister generator with a fixed initial seed of 12,345 for 10,000 iterations. Median and the 5th and 95th percentiles were reported.

Table 1 Information used in modeling cat population dynamics from 1994 to 2005 if no spay/neuter voucher program had been initiated in Santa Clara County, California.

Parameter	Owned cats	Unowned cats	References	
Kittens/litter	4.25	3.6	(Pedersen, 1991; Scott, Levy & Crawford, 2002)	
Kitten mortality rate (%)	30	75	(Jemmett & Evans, 1977; Nutter, Levine & Stoskopf, 2004; Scott, Geissinger & Peltz, 1978)	
Life expectancy (years)	12	4.7	(Levy, Gale & Gale, 2003; New et al., 2004)	
Litters per female per year	2.1	1.4	(Pedersen, 1991; Nutter, Levine & Stoskopf, 2004; Levy, Gale & Gale, 2003; Scott, Levy & Crawford, 2002)	
Percent female	55	45	(Levy, Gale & Gale, 2003)	
Sexually intact (%)	14	94.5	1993 survey	
Surrendered to shelter (%)	3.0	7.3	1993 survey and shelter statistics	

Results

Dog survey results

Twenty nine percent of responding county households reported that they owned dogs (unchanged from 1993); the average household owned 1.9 dogs, representing an increase from 1.3 dogs in 1993. Using US census data led to an estimate of 332,000 owned dogs in Santa Clara County (assuming Palo Alto has the same ownership frequency). Registered and unregistered purebred dogs were 33% and 18% (total = 51%) of the dog population, respectively; the remainder (49%) of dogs was either mixed or unknown breeds. Dogs were acquired from a variety of sources; the most common were friends or relatives (30%), breeders (25%), public or private animal shelter (15%), with the remainder (less than 10% each) coming from a breed rescue group, a newspaper advertisement, found as stray, being born at home, acquired from a pet store, and rare other sources (Fig. 1).

Figure 1 Source of acquisition of dogs from Santa Clara County survey, 2005.

Seventy five percent (75%) of owners reported surgically sterilizing their dogs. Among those that declined to alter them, 28% of owners said this was a deliberate decision, and none claimed that cost was a justification for not sterilizing. Thirty three percent (33%) of the unaltered dogs were intended for breeding purposes, and 17% were puppies too young for surgery. Of the 99 unaltered dogs, 70 (70.7%) were male and 29 (29.3%) were female.

Twenty one households (2%) in the survey acknowledged feeding dogs they did not own, with an average of 2.3 dogs per feeding household. An examination of zip codes indicated that the majority of these dogs were found in the downtown and north and east sides of the city of San Jose. With some exceptions, these areas are in the lower socio-economic range of households in San Jose. Using US census data, this leads to an estimate of approximately 15,650 transiently or permanently stray dogs throughout the county, or 4.7% of the county’s dog population.

Cat survey results

Twenty five percent of households reported owning cats, representing a decrease from 30% in 1993 (p = 0.013). With an average of 1.7 owned cats per household (a figure unchanged since 1993), the county’s owned cat population was estimated at 256,000 cats. Most cats (85%) were characterized as domestic varieties; only 3% were claimed to be registered pedigree (a figure unchanged since 1993), while others were described as unregistered pedigreed or unknown breed. The percentage of cats kept strictly indoors rose from 33% in 1993 to 49% in 2005 (p < 0.001); only 8% were currently described as strictly outdoors, down from 14% in 1993 (p < 0.001).

The most common source of owned cats was from a friend or relative (42% in 2005 versus 33% in 1993), followed by being found as a free-roaming homeless cat (20% in 2005 versus 32% in 1993), a public or private animal shelter (16% in 2005 versus 12% in 1993), a breed rescue group (9% in 2005 versus 2% in 1993), a breeder (4% in both years), an ad in a newspaper or adopted or purchased in a pet store (2% in 2005 versus 6% in 1993), a negligible percentage born at home (<1% in 2005 versus 6% in 1993), and the remainder coming from various minor or unknown sources. The p-value comparing the source distribution of owned cats between 2005 and 1993 was <0.001.

In 2005 most cat owners (92.8%) had their cats surgically sterilized, compared to 86% in 1993 (p < 0.001). Within the 7.2% of cats not spayed or neutered, 48% had owners that deliberately did not want their cats to be sterilized, 13% had owners who wanted to retain the cat for breeding, 13% were kittens, 13% had owners claiming that the costs were prohibitive, and the remainder gave two or more reasons (the most common of which was lack of time to transport the cat for surgery). Thus, only approximately 6% of owned cats were sexually mature and capable of breeding, approximately half of which were female. However, less than one-half of 1% of owners of sterilized female cats allowed their cats to have a litter prior to sterilization.

The rate of reproduction of owned cats in Santa Clara County was 89 cats per 1000 households, in contrast to the higher rate of 112 kittens per 1000 households in the 1996 National Council on Pet Population Study and Policy (Salman et al., 1998). This may be attributable to the high proportion of altered cats in the county (93%) relative to the comparable 1993 figure and the 2005 national average of 86%. In addition, while in 1993 16% of owned cats had a litter prior to altering, in the current study this figure was less than one-half of 1%. While it was beyond the scope of this study to determine the reasons for this change in attitude, it is likely that greater awareness prompted by considerable multimedia public education about the county-sponsored voucher program instituted in 1993 bore at least some responsibility.

When owners were asked about whether their cats were declawed, 8% stated that they were, but 29% of them obtained the cat in that condition. The most common reason given by owners (84%) for declawing was to protect furniture. Owners not electing to declaw their cats protected their furniture through a variety of means, including having scratching posts and mats, using a spray bottle, clipping the claws, applying double-sided tape, and making loud deterrent noises.

Many individuals fed stray cats: 7% of household respondents admitted to feeding an average of 3.2 cats, a decrease from 10% with an average of 3.4 cats in 1993. Relying on U.S. census data, the estimated fed stray cat population is therefore approximately 135,000 cats, or approximately 35% of the total owned and fed free-roaming/unowned cat population in the county (391,000 cats, which represents a drop from 416,000 in 1993). Only 5.5% of these cats were either trapped or taken to be surgically sterilized by their people feeding them. Fifty six percent of the cats were fed daily, while the remaining cats were fed from once every other day to only occasionally. The cats were most commonly fed on the doorstep of a person’s home (62%), followed by an office (15%), a park (12%), and a shopping center (<1%). Fed stray cats were either alone or belonged to colonies ranging in size from 2 to 25 cats. Two-thirds of the fed stray cats were too wild to be picked up and were defined as feral; the remaining third were classified as unowned (although some of these may have had owners unknown to the survey respondent). Forty seven percent of the female stray cats were known to have had at least one litter, which is probably a conservative estimate. Over half of the known litters were allowed to remain free and disperse into their neighborhoods. Of the remaining kittens, half were kept by the feeder, while the others were given away or taken to an animal shelter. Of the females who had litters, 58% were not trapped or taken to a veterinarian after having a litter, remaining free to potentially breed again.

Population changes at Santa Clara County animal shelters

Changes in dog shelter intakes for Santa Clara County (and the Peninsula Humane Society and SPCA shelter in adjacent San Mateo County for comparison) are shown in Fig. 2. Dog intakes declined 13,643 to 8,441 (38.1%) from 1992–2005 in Santa Clara County. An external explanation for the observed trend is supported by the findings in adjacent San Mateo County, where dog intakes declined by a similar 35.7% between 1990 and 2004. These proportions were not significantly different (p = 0.11).

Figure 2 Regression analysis of intake of dogs at shelters in Santa Clara County (r = 0.95) and San Mateo County (r = 0.97) over time (1990–2005).

Figure 3 Secular changes in cat intakes in Santa Clara County, 1982–2007, indexed by historically relevant events.

Figure 4 Regression analysis of intake of cats at shelters in Santa Clara (r = 0.98) and San Mateo Counties (r > 0.99), 1990–2006.

Field services in Santa Clara County ended in 1992; at that time 60% of cats were brought in through field services. Field services resumed in November 1993 in some cities.

A substantially different picture emerged when examining changes in cat shelter intakes in Santa Clara and San Mateo Counties (Figs. 3 and 4). Intakes in Santa Clara County dropped 22,473 to 16,369 (27.2%) from 1993 to 2004 and 22,473 to 16,807 (25.2%) from 1993 to 2005, compared to a drop of 8,252 to 6,078 (26.3%) in San Mateo County from 1993 to 2004. Although the two 1993 to 2004 proportions were similar (p = 0.16), there was an overall decline in annual intakes in Santa Clara County of 6,104 cats to 2004 (509 cats per year) for the 12 year period, compared to 2,174 cats in the same 12 year period for San Mateo County (181 cats per year). The absolute changes are economically more germane to counties with respect to shelter expenses because expenditures are based on the per diem cost of maintaining individual cats. The results of the ARIMA (1,0,1) projections indicated a higher-than-expected cat intake to shelters in Santa Clara County during the years when the voucher program was in effect, i.e., 1994–2005 (Fig. 5). The ARIMA (1,0,1) projections further showed that the observed numbers of cats brought in by the field service did not substantially differ from the expected numbers during the same time period (Fig. 6).

Figure 5 Observed numbers of cats surrendered to the shelters in Santa Clara County versus autoregressive integrated moving average (ARIMA) projected numbers of surrendered cats.

Figure uses the 1982–1993 data (before the launch of the spay/neuter voucher program) shelter data. The lower (LCL) and upper (UCL) 95% confidence limits of the ARIMA projection are also presented.

Figure 6 Observed numbers of cats brought to the shelters in Santa Clara County by the field service versus the autoregressive integrated moving average (ARIMA) projected numbers of cats.

Figure shows cats brought in by field service using the 1982–1993 (before the launch of the spay/neuter voucher program, shelter data). The lower (LCL) and upper (UCL) 95% confidence limits of the ARIMA projection are also presented.

Information provided by the HSSV shelter indicated that the majority of cats entering the shelter were unweaned kittens and feral cats. From 2000 to 2004, the HSSV euthanized 53,419 cats deemed unadoptable: 14,406 were too young (under four weeks of age), 7,912 were unsociable, and 7,595 were feral.

Under the voucher program, 20,419 cats were surgically sterilized from 1994–2001 and an additional 6,231 cats were sterilized from 2001–2003. While the program was initiated at the end of 1994, public interest did not start until mid-1995, when a local television station and newspaper ran a story about it.

The San Jose program was initially free to the public; however, various program changes over time were instituted. Veterinarians were reimbursed at a set fee of $25 female and $15 male. Pregnancies could add to the veterinarian reimbursement up to $50, and anatomical issues adjusted the price to as high as $150. In 1996 modifications included requiring a $5 co-pay, and a requirement that cat owners obtain a $5 license and rabies inoculation. While these changes increased the veterinarian reimbursement, they also created a negative effect on the program, as voucher requests declined from 5,600 in the first 16 months of the program to only 2,800 for the year following the changes.

The San Jose voucher program ended in 2003, but the county program continued. Utilizing assumptions in Table 1, if no voucher program had been initiated, the same cats enrolled in the voucher program (assuming that 65% were owned, based on the 2005 survey results) would have produced approximately 312,000 kittens between 1994 and 2005, and approximately 8,600 additional cats would have entered (6,200 surrendered and 2,500 brought in by the field service) the shelters in Santa Clara County. This would have incurred an additional cost of approximately $2.15 million, with the HSSV charge to cities for stray cat services under their contract cost of $250 per cat. If the cost per cat for spay/neuter surgery in 2001–2002 ($23.21 average for all surgeries) can be assumed to be constant from 1994–2005, then the expected cost of the HSSV voucher program was approximately $620,000. Thus, the net gain of the program from reducing the number of cat shelter intakes was approximately $1.53 million. Not counted would be the added burden of approximately 44,000 cats to the stray population in the county.

The proportion of feral cats actually altered in the program considerably fluctuated: from 77% in 2006 to 82% in 2007 to 48% in 2008 (the latter data is from San Jose only). Table 2 contains projections of how county cat and shelter populations would be expected to change in the absence of the voucher program under different owned versus feral cat ratios. Under all plausible scenarios, ranging from 20% to 80% of the altered cats being feral, the costs to the shelters would have likely exceeded $2 million over the 12-year life of the program, and at the higher proportion of feral cats the costs would have likely exceeded $6 million.

Table 2 Projected impact of hypothetical absence of the 12 year spay/neuter program on cat populations, shelter intake, and municipal cost in Santa Clara County.

Median and the 5th and 95th percentiles (in parentheses) are reported (×$1,000).

Percentage of surgeries performed on owned cats	Additional number of owned cats	Additional number of stray cats	Cats voluntarily surrendered to shelter	Cats brought in by field service	Total additional shelter cat intake	Cost to shelter for additional surrendered and stray cats	
20	193	265	6	20	25	$6,333	
	(110, 388)	131, 593)	(3, 12)	(10, 44)	(13, 55)	(3,242, 13,817)	
30	226	181	7	13	20	$5,034	
	(140, 412)	(98, 367)	(4, 13)	(7, 27)	(11, 40)	(2,867, 9,893)	
40	239	124	7	9	16	$4,078	
	(159, 398)	(73, 226)	(5, 12)	(5, 17)	(10, 28)	(2,549, 7,089)	
50	242	84	7	6	13	$3,351	
	(173, 368)	(54, 138)	(5, 11)	(4, 10)	(9, 21)	(2,282, 5,291)	
60	241	55	7	4	11	$2,823	
	(180, 342)	(38, 85)	(5, 10)	(3, 6)	(8, 17)	(2,035, 4,135)	
70	236	35	7	3	10	$2,411	
	(182, 319)	(25, 50)	(5, 10)	(2, 4)	(7, 13)	(1,814, 3,324)	
80	231	20	7	1	8	$2,099	
	(182, 303)	(15, 27)	(5, 9)	(1, 2)	(6, 11)	(1,625, 2,792)	

Discussion

This study documents the positive impacts publically subsidized low-cost spay and neuter programs can have that often go unmet in communities: pet population control, leading to the prevention of the proliferation of feral dog and cat populations, slowing the flow of animals into shelters both voluntarily and through field services, and reduction in the incidence of humane destruction of animals. But they also extend to other issues of economic importance to communities; namely, reduction in capital and ongoing animal control expenditures that come under a municipality’s jurisdiction. In contrast, the implication of cessation of such programs can be seen in Figs. 2 and 3 when the decline in shelter admissions of dogs and cats became attenuated. The problem could be exacerbated over time as the human (and hence pet-owning) population increases.

The finding that there were over 15,000 dogs (4.7% of the county’s dog population) estimated to be transiently or permanently stray throughout the county is troubling from societal and public health standpoints. The absence of a domestic environment can lead stray dogs, which are by nature gregarious, to form packs that can become aggressive and endanger other animals and even humans. The origin of such a large number of dogs is worthy of further research, as this study did not explore whether these were free-roaming dogs or those kept in temporary foster or rescue care.

Specific breed information was not available for dogs in Santa Clara County animal shelters. Respondents claimed 51% of their dogs were registered and unregistered purebred dogs. This stands in contrast to a 1996 national survey that found 30% of dogs relinquished to shelters were purebred (Salman et al., 1998), and the Humane Society of the United States estimates that 25%–30% of shelter dogs are purebred (The Humane Society of the United States, 2011). Nationally, purebred dogs are substantially less likely to be relinquished to animal shelters than dogs of mixed breed (Salman et al., 1998).

The dogs with greatest likelihood of successful adoption from county animal shelters are puppies (Lepper, Kass & Hart, 2002). By the time dogs reach the age of one year, though, their risk of unsuccessful adoption following relinquishment rises considerably; again, particularly true in pit bull-like breeds (Lepper, Kass & Hart, 2002). Aggressive dog behavior is a major reason dogs are euthanized at the county shelters (Kass et al., 2001). To reduce dog intakes, municipalities should consider how the establishment of free or low cost puppy training programs (potentially mandatory for shelter adoptions) might impact shelter populations. A collaborative effort among multiple community agencies, including animal control, non-profits, and local pet industry businesses should be explored.

Another important finding is the enumeration of the substantial unowned cat population in Santa Clara County, two-thirds of which are feral. Also notable is that the majority of unowned cats entering the animal shelters in the study were arguably unsuitable for adoption, with over 50% being feral or unweaned kittens. Such cats are often quickly euthanized. Preventing such input defies simplistic solutions, because although 93% of cat owners were willing to have their own pets surgically sterilized, it is unrealistic to expect the 7% of the population that feeds an average of 3 stray cats to assume the hundreds of dollars necessary to surgically alter these cats.

Conversely, the cost of not altering the cats is to add 3.5 kittens per year (Nutter, Levine & Stoskopf, 2004; Pedersen, 1991) for each stray female, which at the cost to a shelter of approximately $250 per cat would cost a shelter almost $900 in husbandry expenses for those 3.5 kittens; were they not sheltered, the kittens would be expected to have 75% mortality (Table 1). The underscores why low-cost spay and neuter programs directed to reducing the un-owned and feral cat populations continue to be integral to not only reducing cat mortality at the shelters, but also to managing the cost to the various municipalities to handle and house the stray cats. Santa Clara County’s contribution of $45 to alter a stray cat under its separate feral spay/neuter program created an immediate savings of over $200 for just the first litter that permanently results in non-reproducing cats. The county program also subsidized shots, and for a time, FELV testing. The earlier such cats can be sterilized, the greater the potential savings to municipalities. The savings would be expected to grow over additional years. Moreover, under all plausible scenarios shown in Table 2 the voucher program would have resulted in a net savings in expenditure.

If stray cat-feeding citizens can be convinced through public education to avail themselves of population control options by making them more affordable and they are provided with instructions and resources as to how to accomplish this activity, the savings in costs and lives will be substantial. This study shows approximately 93% of county residents did not make an effort to sterilize unowned cats. Only 5.5% of the unowned but fed cats were surgically sterilized. Efforts should be focused on removing barriers and finding ways to encourage those who feed free-roaming cats to take this important step. Because the study shows that 62% of unowned but fed cats are fed in people’s yards, efforts should be intensified to sterilize cats living in close proximity to homes, rather than less accessible colonies.

This study’s limitations include the assumption that the participating individuals are representative of the county’s population. Interviews were conducted via telephone, with the non-telephone-owning segment of the county excluded by design, and to the extent that this subgroup differs in their pet ownership and practices the findings cannot be generalized to them. However, calls were made to each zip code in the county, and the number of respondents completing the survey in each zip code, were proportionate to their share of the county population. Although the finding that shelter intake declined in association with the inception of the voucher program, the presence of extraneous (confounding) factors associated both with time and shelter intake cannot be ruled out, including migration into and out of the county (although the human population actually increased during the study period). At the time of the study, two additional smaller shelters existed in the county: a county facility in San Martin, which served the 5% of the population not living within cities, and a city facility in Palo Alto, which only served Palo Alto residents; these shelters were not expected to have any meaningful impact on intake changes in the shelters in this study.

In conclusion, this study demonstrates the financial and societal value of instituting a low-cost voucher program on a county-wide scale. Although the parameters utilized in the projections and models in this research (e.g., fecundity and mortality) will vary, perhaps substantially, from county to county, they are realistic and based on published observations. It is therefore likely that the qualitative – if not the precise quantitative – benefits of the voucher program in Santa Clara County will be of significance if incepted elsewhere.

We are grateful to Beth Ward and Chris Benninger at the Humane Society of Silicon Valley, Greg VanWassenhove at County of Santa Clara, and Jon Cicirelli at the City of San Jose for providing annual shelter statistics.

Additional Information and Declarations

Competing Interests

Author Contributions

4 Scientific Telephone Samples, Foothill Ranch, California.

The authors do not have any competing interests in this manuscript. Philip Kass is an Academic Editor for PeerJ.

Philip H. Kass and Hsin-Yi Weng analyzed the data, wrote the paper.

Karen L. Johnson wrote the paper.

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
