# Peer review of "Evaluation of animal control measures on pet demographics in Santa Clara County, California, 1993–2006"

_PeerJ, doi:10.7717/peerj.18_

## Round 0.1 · original submission · Major Revisions

Please pay particular attention to the references needed in the introduction, as mentioned by reviewer 1 and the statistical concerns of reviewer 2.

·

Basic reporting

Every statement of fact in the introduction needs to be backed up with a reference; there are no cited references present in the introduction.
L-44 which nation has done quasi-national studies?
L-57 to59 is it needed
L-59 to 60 not sure what this means, needs clarification or is it needed at all?.
L61-108 needs to be rewritten and more concise
L108 should this be “pet population”?

Experimental design

L117 is a bit vague, 7000 numbers or calls, how many refusals, hang Ups no answers?
L118 what entry data was obtained, how did the authors get it?
L129 what analysis did you do with the survey, if descriptive that needs to be stated
Was sample size justified?
L129 sentence not needed at this point. The mathematical model should be given in way more detail, at this point I could not reproduce this study if I wanted to do so.

Validity of the findings

L149 -150 are the populations comparable, what are the CI,
L147 how did you use US census data? This should be declared in the methods
L167 again how was the census data used?
Three Paragraphs starting from L171 onwards: where did these p values come from, this is not described in the methods, why were these not done for dogs?
L212 again US census data is being used, but there is no way of knowing how this was done.
L118 there needs to be far more info on the questionnaire given the level of detail in the results.
L233 this belongs in the discussion.
L252 where is all this information coming from, again this is not mentioned in the methods.
L 256 same as last comment
L264 which cats are these cats?
L276 there seems to be a mix of results, discussion and new raw data introduced in this paragraph, it is unclear what is going on here.
Paragraph from 291. Unfortunately, while the statements in this paragraph may be true I’m not certain the study shows this at this point, there have been quite a few assumptions made in coming to this point.
Some of the statements in the discussion require being backed up by citations.
Paragraphs starting L 318 and 335 is this line from this work or from some where else, it is not presented in the results

The limitations are not really addressed in Paragraph starting L354. What about other shelters in the area that are not included in the study, perhaps the intake in these has had an effect on the intake in the ones studied? There seems to be no account of migration into and out of the study area, was this assumption made in the mathematical model. Also the figures been used are from external data and not nesecarily from a similar population. There is no mention of the AVMA sourcebook and how the population in this survey is different and why this is so.

Additional comments

This is a really excelent idea but the report on the study has let it down quite a bit. It is difficult to tell how well the study has been carried out as the information is absent in the materials and methods. There seems to be 3 major parts to the study and this has made it very difficult to track throughout the paper.
I think that it would be of benefit to consider turning this into more than one report. Perhaps one report on the Telephone survey as it seems not to be used to great effect in the moddeling and then combined shelter/modelling information.
I would also consider it beneficial to contact other shelters in the Santa Clara area (especially new ones) as these could have an effect on the shelters being studied.

Reviewer 2 ·

Basic reporting

The article is well organized and conforms to professional standards of expression. Sufficient introduction and background are included in the article. Figures are appropriately described and labeled. All results relevant to the hypothesis are included.

Experimental design

In this article, the authors used a prospective cross-sectional study of 1000 households to evaluate characteristics of the owned and unowned dogs and cats in Santa Clara County, California. This study is novel. Research questions are clearly defined. Methods are clearly described.

Validity of the findings

The authors used the numbers of owned and unowned dogs and cats in 1993 and 2005 to support their conclusions about the effectiveness of the cat spay/neuter program. During these years, Santa Clara County established a spay/neuter program for cats; but San Mateo County didn’t have such a program. The authors compared the yearly animal shelter cat intakes between the two counties. Time series analysis showed a greater than expected decline in the number of cats surrendered to shelters in Santa Clara County. However, the comparison between two counties didn’t provide convincing evidences to support the effectiveness of the spay/neuter program.

1. Misleading is the authors’ claim that “Dog intakes declined from 1992-2005, as they similarly did for an adjacent county (San Mateo). However, car intakes declined significantly more in Santa Clara County than San Mateo.” In fact, authors’ data showed that the two counties have a larger percentage difference in the reeducation of dog intakes than cat intakes. Dog intakes declined 38.1% in Santa Clara and 35.7% in San Mateo (2.4% difference), whereas cat intakes declined 27.2% in Santa Clara and 26.3% in San Mateo (0.9% difference). The authors should revise these sentences to provide a more subjective conclusion.

2. The authors should provide citations to support their claim that “the greatest threat to the lives of cats … from the threat of being unowned or becoming unwanted”.

3. On page 8, since the households feeding unowned dogs are in lower socio-economic area, the authors should use the number of households in the lower socio-economic area to calculate the total number of stray dogs.

4. On page 11, the authors should calculate a P value to compare the reeducation of cat intakes between Santa Clara to San Mateo, similar to the P value that they calculated for dog intakes.

5. The authors should conduct the time series analysis on the number of cat intakes instead of the number of surrendered cat, since the number of cat intakes is more relevant to the cost of cat shelter service.

6. On page 16, the authors should consider the 75% mortality rate of the kittens of stray cats in the authors’ estimation of ‘add 3.5 kittens per year’.

7. On page 16, it should be “62% of unowned but fed cats are fed in people’s yards”, instead of “62% of stray cats”.

---

## Round 0.2 · accepted · Accept

My only additional suggestion is that a sentence explaining the major aim of the study should be added at the beginning of the abstract. As it is written, it seems the aim is to survey households. Perhaps a sentence or two at the beginning of the abstract stating that "The measurable benefits of animal control programs are unknown and the aim of this study was to determine the impact of these programs on pet population changes." should be added.